# Loss of Function of the RRMF Domain in OsROS1a Causes Sterility in Rice (*Oryza sativa* L.)

**DOI:** 10.3390/ijms231911349

**Published:** 2022-09-26

**Authors:** Jian-Hong Xu, Faiza Irshad, Yan Yan, Chao Li

**Affiliations:** 1Hainan Institute, Zhejiang University, Sanya 572025, China; 2Zhejiang Key Laboratory of Crop Germplasm, College of Agriculture & Biotechnology, Zhejiang University, Hangzhou 310058, China; 3Shandong (Linyi) Institute of Modern Agriculture, Zhejiang University, Linyi 276000, China

**Keywords:** *ROS1*, DNA demethylation, CRISPR/Cas9, pollen fertility, bisulfite sequencing, rice (*Oryza satia* L.)

## Abstract

For crop seed production, the development of anthers and male fertility are the main agronomic traits and key biological processes for flowering plants. Active DNA demethylation regulates many plant developmental processes and is ensured by 5-meC DNA glycosylase enzymes. To find out the role of *OsROS1a*, *OsROS1a* gene editing mutants were generated using the CRISPR/Cas9 system. The *osros1a* mutants had shrink spikelets, smaller anthers and pollen grains, and were not stained by iodine staining showing a significant reduction in total soluble sugar and starch contents as compared to wildtype (WT), which caused complete male sterility. Similarly, the expression of genes involved in pollen and anther development was decreased in *osros1a* mutants as compared to WT. Furthermore, bisulfite sequencing showed that the CG and CHG methylation of the *OsPKS2* gene promoter was significantly increased in the *osros1a* mutant, which caused a reduced expression of *OsPKS2* in *osros1a* mutants. DNA methylation of the *TDR* gene promoter was similar between WT and *osros1a* mutants, indicating that the DNA methylation effect by *OsROS1a* was gene specific. The expression of *O**sROS1a* in the mutants was not changed, but it produced a frame-shift mutation to truncate the Pem-CXXC and RRMF domains. Combined with previous studies, our findings suggested that the RRMF domain in OsROS1a is the functional domain and loss of RRMF for *OsROS1a* causes sterility in rice.

## 1. Introduction

DNA methylation plays a crucial role in plant growth, development, and stress responses [1]. In plants, DNA methylation usually occurs at the cytosine (C) base in CG, CHG and CHH configurations (H = A, C, or T), and is dynamically regulated by balanced methylation and demethylation [1]. Plant 5-meC DNA glycosylases are placed in the DEMETER-like (DML) family, which is related to the HhH-GPD superfamily and is the main functionally-varied group of DNA glycosylases [2]. DML proteins ranging from 1100 to over 2000 residues are unusually large DNA glycosylases, which are bifunctional enzymes having both DNA glycosylase and apurinic/apyrimidinic (AP) lyase activities [3]. Specific DNA glycosylases, including Repressor of silencing 1 (ROS1) [4], DEMETER (DME) [5], DML2 and DML3 [6] catalyze the cytosine demethylation in *Arabidopsis*. DNA demethylation mediated by DME is important for the reproduction of plants, and the inheritance of loss-of-function paternal or maternal mutant *dme* alleles resulting in a reduced sperm transmission or seed abortion, respectively [5,7]. present only in dicots, but not in monocots. ROS1 encodes a DNA glycosylase/lyase that represses DNA methylation of promoter DNA [8]. The loss-of-function of *ROS1* causes DNA hypermethylation of CG and CHH and decreases the related gene expression in *Arabidopsis* [3,4]. The rice genome contains four *ROS1* paralog genes (*Os**ROS1a, OsROS1b*, *OsROS1c,* and *Os**ROS1d*) that mediate DNA demethylation [9]. *OsROS1a* is highly expressed in rice anthers and pistils [10]. The disrupted *OsROS1a* mutants exhibited severe defects in both male and female gametogenesis to produce a sterile phenotype [10,11]. The rice vegetative cell genome is actively demethylated by *OsROS1a* and is vital for viable seed production, and sperm non-CG methylation is indirectly promoted by DNA methylation in the vegetative cell, which suggests that dynamic DNA methylation reprogramming occurs during plant embryogenesis [12]. The aleurone layer is the most nutritious part of cereal grains which stores lipids, vitamins, proteins, and minerals. The point mutation in the fourteenth intron of *OsROS1* generates a new transcript *mOsROS1* with a 21-nt insertion, which can lead to DNA hypermethylation and suppresses the expression of two transcription factors, *RISBZ1* and *RPBF*, increasing the number of aleurone cell layers [13,14]. Knockout or knockdown of *O**s**ROS1b* causes DNA hypermethylation of *Tos17* retrotransposons in rice, and overexpression of *O**s**ROS1b* extensively reduces DNA methylation of the rice genome [15]. Furthermore, *ROS1* is involved in the seed development of rice, wheat and barley by its epigenetic influence on the accumulation of seed storage proteins (SSPs) [15,16].

Pollen fertility is important for successful seed production in flowering plants. However, defective development of the anthers can lead to either an absence or formation of non-functional pollen grains [17]. Successful development of male reproductive organs includes events such as specification of the meristem, cell differentiation, cell-to-cell communication, meiosis and mitosis [18,19,20,21]. The development of pollen is controlled specifically by four sporophytic cell layers of the anther (tapetum, middle layer, endothecium and epidermis), which surround the gametophytic pollen grains [22]. Many genes involved in the development of pollen and anthers have been identified in rice. A plant-specific type III polyketide synthase gene *OsPKS2* is involved in the normal development of pollen wall formation and mutation in the *OsPKS2* caused male sterility in rice [23]. The rice tapetum degeneration retardation gene (*TDR*) has an essential role in regulating the transcriptional network for the development and degradation of tapetum, and mutation in this gene caused male sterility [24,25]. MEIOSIS ARRESTED AT LEPTOTENE1 (*MEL1*) gene has a crucial role in microsporogenesis, abnormal accumulation of *MEL1* can make a semi-sterile phenotype in rice [26]. Cytochrome P450 family member *CYP704B2* is specifically expressed in the tapetum and microspores, which is required for the synthesis of cutin monomers, and its mutant *cyp704B2* shows a male sterile phenotype with a swollen sporophytic tapetal layer and aborted pollen grains [27]. Pollen development, being extremely susceptible to the cellular environment, is regulated by the successive expression of genes specifically expressed in reproductive tissues. Male gametogenesis also shows abnormalities and becomes inactive if any imbalances occur in the expression of anther and microspores development-related genes.

This study aimed to reveal the underlying mechanism of how *OsROS1a* regulates pollen fertility. The CRISPR/Cas9 system was used to generate *OsROS1a* gene editing mutants, and the *osros1a* S6, S7, and S16 mutants exhibited complete male sterile, which resulted from the disrupted Pem-CXXC and RNA recognition motif fold (RRMF) domains. The qRT-PCR results showed that the expression of genes involved in pollen and anther development, and starch biosynthesis decreased in these mutants as compared to WT. Furthermore, bisulfite sequencing showed that the CG and CHG methylation of the *OsPKS2* gene promoter was significantly increased in the *osros1a* mutants. These findings could provide new insights to reveal the DNA demethylation of *OsROS1a* on sterility in rice.

## 2. Results

### 2.1. Phylogenetic Analysis of OsROS1a Glycosylase Domain

*Os**ROS1a* contains 17 exons that encode a protein having 1952 amino acids (aa), and 3′-and 5′-UTRs of 607 and 73 bp, respectively (Figure 1A). The *OsROS1a* contains three key domains of DNA glycosylase, Per-CXXC and RRMF (Figure 1B), having lengths of 141, 31 and 102 aa, respectively. The location of these domains in the OsROS1a CDS protein sequences was 1472–1612 for DNA glycosylase, 1797–1827 for Per-CXXC, and 1831–1932 for RRMF (Figure 1C).

To determine the evolutionary relationship of *OsROS1a* genes in different species, its protein sequences were used, and 159 homologous gene copies were identified in 52 plant species. The conserved DNA glycosylase domain (151 amino acids) of 159 genes was then used for constructing the phylogenetic tree with the neighbor-joining (NJ) method (Figure 2). The phylogenetic tree can be clustered into 4 clades, monocots *ROS1* (mROS1), dicots *DME* (dDME), dicots *ROS1* (dROS1) and dicots *DML2* (dDML2) (Figure 2). Alignment of multiple sequences of deduced amino acid sequences revealed that all homologous *ROS1* genes contain the DNA glycosylase domain. The evolutionary analysis discovered that DNA demethylase gene copy number varied greatly among different species or different families of the same species. The homologous copy number varies from 1 to 6 in different plants.

The sequence logos of the identified DNA glycosylase domain in all 52 species were generated using the WebLogo program to further verify the conservation of aa residues (Figure 3A). A total of three conserved motifs were determined, including the Helix-hairpin-Helix (H-h-H), GPD and [4Fe-4S] motifs. The motifs length ranged from 21 to 26 aa (Figure 3B). The [4Fe-4S] cluster motif contained four cysteine residues that function to keep a [4Fe-4S] cluster and was essential for 5mC excision.

### 2.2. Generation of OsROS1a Gene Editing Mutants by CRISPR/Cas9

Previous studies showed that the disrupted *OsROS1a* exhibited severe defects in both male and female gametogenesis to produce a sterile phenotype [10,11]. To well describe the function of *Os**ROS1a* in active DNA demethylation on pollen and seed development, the stable *Os**ROS1a* gene editing mutants were created using the CRISPR/Cas9 system, and three stable mutants (S6, S7, S16) were obtained. The S6 and S16 mutants are homozygous mutations on both target sides, while S16 had a homozygous mutation in the first target and a biallelic mutation in the second target (Figure 4A and Appendix A). All mutants caused a frame shift and generated premature stop codons at the thirteenth exon of *OsROS1a* (Figure 4B), which made a frame shift from the 1798th amino acid. The S6 and S16 mutants created a stop codon at the 1807th amino acid, while S7 terminated translation by a stop codon at the 1820th amino acid by a one base deletion in the first target (Figure 4B). Both frame shifts occurred before Pem-CXXC and RRMF domains that altered the function of cytosine DNA demethylation.

### 2.3. The osros1a Mutations Cause Complete Male Sterility

All three obtained *osros1a* mutants were completely male sterile, exhibiting shrunk spikelet morphology and smaller anthers compared to WT controls (Figure 5A–C). To understand the male sterile phenotype of *osros1a* mutants, we then examined the starch-staining capacity of pollen grains and found that the pollen grains of the mutants were not stained by I_2_-KI solution, indicating that they contained little or no starch. While the pollen grains were stained dark within WT controls (Figure 5D–F), and the fertile seed percentage in all mutants were 0% compared to WT controls (77%) (Figure 5G). Furthermore, at the harvest stage, agronomic traits were examined and no significant difference was found between the WT and *osros1a* mutants (Appendix A). The plant height only increased by 2.3% and 0.4% in S6 and S16 mutants, respectively, when compared to WT controls. Similarly, the primary branch number per panicle increased in S6 and S16 by 2.7% and 5.4% compared to WT controls. While in the case of panicle length, a 2.9% decrease was observed only in S16 compared to WT controls. Moreover, the tiller number decreased by 1.3% in S16 mutants only compared to WT (Appendix A).

To further investigate the cellular defects in the *osros1a* mutants, we performed transverse section analysis on the anthers of the WT and *osros1a* mutant. Previous studies have divided the development of rice anthers into 14 stages based on the morphological landmarks of cellular events [28,29]. The anther sections were observed at developmental stages 6, 9, 10 and 13 under light microscopy. At stage 6 normal epidermis, endothecium, middle layer, tapetum, and microspore mother cells (MMC) were found in both WT and the *osros1a* S16 mutant (Figure 6A,E). At stage 9, the microspores were released from tetrads, tapetal cells had deeply stained cytoplasm and the middle layer was hardly visible, and the middle layer appeared degenerated and almost invisible in both mutant and WT. However, the microspores of the WT were globular, whereas those of the *osros1a* mutant were irregularly shaped (Figure 6B,F). At stage 10, in contrast to the WT microspores that were vacuolated and round, the *osros1a* mutant microspores appeared degraded and irregularly shaped (Figure 6C,G). At stage 13, the WT anther locule was full of mature pollen grains with completely formed pollen walls and starch accumulation, which can be deeply stained with toluidine blue. By contrast, the pollen grains of the *osros1a* mutant were aborted (Figure 6D,H), indicating that pollen viability and function are mainly affected by the accumulation of starch and lipids [28,30].

Sugar supply and starch accumulation are essential for pollen grain development that will further affect seed maturation [31,32,33]. Therefore, the total soluble sugar and starch contents were analyzed in the anthers to investigate whether the *OsROS**1a* gene can affect sugar and starch synthesis in pollen grains. The reduction of 36.2% and 56.5% soluble sugar was observed in S6 and S16 mutants, respectively, compared to WT controls (Figure 7A), and similarly 37.1% and 57.9% starch content was significantly (*p* < 0.01) decreased in both mutants, respectively, compared to WT controls (Figure 7B) This was consistent with the pollen staining and transverse section analysis (Figure 5 and Figure 6), confirming that the *osros1a* mutation caused complete male sterile in rice.

### 2.4. The Reduction Expression of Genes Involved in Anther and Pollen Development in Osros1a Mutants

To identify the potential targets responsible for the *osros1a* mutant phenotype, qRT-PCR was performed to validate the expression of anther and pollen development-related genes in young panicles. The results showed that the expression of *OsPKS2* and *CYP704B2* genes was reduced in S6 and S16 mutants compared to WT controls (Figure 8B,D). It might be suggested that *OsROS1a* may involve in the activation of these genes by the process of DNA demethylation. Interestingly, the *OsROS1a* gene in mutants had half the expression level of WT controls (Figure 8A), suggesting that the frame-shift mutations that altered the Pem-CXXC and RRMF domains could cause the sterility phenotype in rice. The expression of genes related to soluble sugar and starch synthesis were further investigated to validate the previously obtained soluble sugar and starch content analysis results and found that the expression of *CSA* was decreased in mutants compared to WT controls (Figure 8G), and the expression of other genes was not significantly changed (Figure 8C,E,F,H) These results are consistent with the pollen staining and transverse section analysis (Figure 5 and Figure 6), showing that the *os**ros1a* mutant promotes the pollen sterility phenotype.

### 2.5. DNA Hypermethylation of OsPKS2 Gene Promoter in Osros1a Mutants

To reveal the function of *OsROS1a* in DNA demethylation in pollen, the DNA methylation level of the 221 bp *OsPKS2* gene promoter was investigated (Figure 9A). The methylation levels at both CG and CHG sites of the *OsPKS2* promoter were increased in *osros1a* mutants compared to WT controls (Figure 9B,C). In CG residues, 96.15% and 100% methylation levels were observed in S6 and S16 mutants, respectively, while only 51.79% in WT controls. In CHG residues, methylation levels increased from 54.29% in WT controls to 66.15% and 67.69% in S6 and S16 mutants, respectively. However, no obvious difference was observed in CHH residues between WT and S16, whilst a decrease was observed in S6 mutants (Figure 9B,C). These results suggest that DNA hypermethylation in the promoter region of *OsPKS2* may repress the gene expression. While DNA methylation of the *TDR* promoter was not significantly changed between WT and *osros1a* mutants in all three contexts (Appendix A).

## 3. Discussion

The loss function of the *OsROS1a* gene caused complete male sterility with degenerated pollen (Figure 5 and Figure 6), which was consistent with previous studies [10,11,12]. Starch is the key storage reservoir in matured pollen grains in cereal crops, which is important for pollen germination, pollen tube growth, supplying energy and a carbon skeleton. In pollen grains, the starch granule accumulation starts at stage 11 and prolongs throughout stage 12 until maturation [29]. *OsBZR1* gene was expressed in anther vascular tissue, tapetum and the developing seeds, and *OsBZR1* directly promotes *CSA* expression thus activating downstream gene expression [30]. Similar to sugar partitioning in rice, *CSA* is a key transcriptional regulator during male reproductive development, and mutation in the *CSA* results in carbohydrate level reduction in the later anthers and male sterility [33]. The deficiency of starch synthesis in pollen grains could cause male sterility [34,35,36,37]. Our study showed that the soluble sugar and starch contents were significantly reduced in pollen from *osros1a* mutants compared to WT controls (Figure 7), which could be the reason for male sterility affected by disrupted *OsROS1a* gene function.

Previous research has revealed that *OsPKS2* [23], *TDR* [25], *MEL1* [26], and *CYP704B2* [27] are involved in anther and pollen development and mutation in these genes caused rice male sterility. The expression of *OsPKS2* and *CYP704B2* genes were reduced in *osros1a* mutant anther compared to WT controls (Figure 8), suggesting that *OsROS1a* may be involved in the activation of these genes, loss-of-function of *OsROS1a* could degenerate the tapetum and form abnormal pollen walls causing male sterility (Figure 5 and Figure 6). The DNA demethylase genes contain a conserved DNA glycosylase domain, which can be classified into monofunctional and bifunctional types. DNA glycosylase is bifunctional in plants removing the 5-meC base and then cleaving the DNA backbone at the abasic site [38]. ROS1 is a bifunctional DNA glycosylase that can execute DNA demethylation, especially gene promoter DNA methylation by a base excision repair pathway [8,38]. The reduced expression of pollen and anther development, and starch- and soluble sugar-related genes in *osros1a* mutants demonstrated that *OsROS1a* may regulate the activation of these genes by controlling their DNA methylation. Bisulfite sequencing results showed that the CG and CHG methylations in the *OsPKS2* gene promoter were significantly increased in the *osros1a* mutants when compared to WT controls, but not CHH methylation (Figure 6), suggesting that *OsROS1a* facilitated CG and CHG demethylation. However, DNA methylation of the *TDR* promoter was not significantly changed between WT and *osros1a* mutants (Appendix A), indicating that the demethylation action of the *OsROS1* gene on pollen and anther development-related genes was not global, but gene-specific, which was also found in SSP genes in maize [39].

The expression of the *OsROS1a* gene in the mutants was half that of WT controls (Figure 8), and 1807, 1807 and 1820 amino acids were predicted for S6, S16 and S7, respectively, which resulted in the frame-shift mutation to truncate the Pem-CXXC and RRMF domains (Figure 1 and Figure 3). The whole DNA glycosylase domain, followed by an EndIII_4Fe-4S domain and two putative nuclear localization signals are complete [15]. The C-terminal of DME and ROS1 including divergent, circularly Per-CXXC and RRMF domains are conserved [40], which has a function in excising 5-meC in vitro [5,41,42]. The zinc finger CxxC (ZF-CxxC) in DNA methyltransferase 1 (*DNMT1*) can block the catalytic activity of *DNMT1* specifically on non-methylated DNA [43]. Furthermore, the *OsROS1a* knock-in mutant disrupted the whole gene affecting both male and female gametophytes [10], while truncation the RRMF domain in *OsROS1a* knock-out mutants using CRISPR/Cas9 induced pollen and embryo sac defects in rice [11]. Recently, a 75 bp deletion caused the complete loss of the Per-CXXC domain, but retention of the whole RRMF domain, which showed partially sterile pollen in rice [44]. All these results suggest that the RRMF domain in OsROS1a could be the main factor in the sterility in rice. The RNA-Seq and whole-genome bisulfite sequencing will be the future work, which will clearly clarify the molecular mechanism of how the *OsROS1a* gene affects the sterility phenotype in rice through its DNA demethylation function.

## 4. Materials and Methods

### 4.1. Identification of OsROS1a Homologous Genes and Phylogenetic Analysis

The gene structure (Exon-intron distribution) and protein (CDS) sequence of the DNA demethylase gene *OsROS1a* was obtained from Phytozome (https://phytozome-next.jgi.doe.gov/) (accessed on 19 October 2018). The conserved domain sequence position of the OsROS1a protein was searched using the Inter Pro software. The protein sequences of OsROS1a were used to identify the homologous gene copies in 52 plant species by BLAST search. The protein sequences of all identified *Os**ROS1a* homologous genes were aligned via Muscle by the default settings with manual alterations. The phylogenetic analysis was done by the neighbor-joining (NJ) method using MAGA-X with the default parameters, and the bootstrap test with 1000 replicates was made to determine the confidence of the evolutionary tree [45]. The conserved DNA glycosylase domains (151 amino acid sequences) were analyzed by the WebLogo program (http://weblogo.berkeley.edu/) (accessed on 15 January 2019).

### 4.2. OsROS1a Gene Editing Mutants Generated by CRISPR/Cas9

The CRISPR/Cas9 system was used for generating *osros1a* mutants [46]. Two single guide RNAs (sgRNAs) were designed to precisely target the thirteenth exon of the *OsROS1a* gene by using the web tool (http://cbi.hzau.edu.cn/crispr/) (accessed on 19 December 2016), which were ligated with rice OsU6a and OsU6b small nuclear RNA promoters, respectively. The sgRNAs were cloned into a binary vector that contained the sgRNA and Cas9 expression cassettes. Then the binary vector was transferred into rice (Nipponbare) calluses by following the method of *Agrobacterium*-mediated transformation of *Japonica* Rice.

Genomic DNA was isolated from the leaves of the rice transgenic plants to detect the mutation types. The Cas9-specific primers were used to identify the successful transgenic plants by PCR and Appendix A showed the gel electrophoresis image. And the target gene-specific primers were used to amplify and sequence the DNA fragments containing the target sequences (Appendix A). Then the DSDecode (http://skl.scau.edu.cn/dsdecode/) (accessed on 12 September 2017), a web-based tool, was used to further analyze the genotypes of the targeted mutations.

### 4.3. Pollen Fertility Examination and Histochemical Assay

Floral organs were photographed with a dissecting microscope. The anthers from the wild type (WT) and *osros1a* mutants were sampled from the spikelets just before flowering and a 1% potassium iodide (I_2_-KI) solution was used to stain the pollen grains. A Nikon eclipse Ni fluorescence microscope was used to visualize and photograph the stained pollen grains. For transverse section analysis, spikelets of various anther development stages according to the previous study [29] were collected and frozen in OCT compound (Tissue-Tek) (Sakura Finetek, Torrance, CA USA). Samples were sectioned with the help of a freezing microtome (Shandon Cryotome FE) (ThermoFisher Scientific, Waltham, MA USA). Then, 0.05% toluidine blue dye was used to stain the crossed sections and microscopic images were captured by a Nikon eclipse Ni fluorescence microscope.

### 4.4. Total Soluble Sugar and Starch Contents Analysis

The 0.05 g anthers from WT and *osros1a* mutants were weighed and ground in 5 mL of 80% ethanol according to the Anthrone method [47]. The mixed samples were centrifuged at 8000 rpm and 1 mL supernatant was obtained. After centrifugation, the pellets were used for starch analysis. The 5 mL anthrone was mixed with 1 mL supernatant and heated at 100 °C in a water bath for 10 min. The mixture was cool down, and the sugar content was then determined by a spectrophotometer at a wavelength of 620 nm. Starch analysis was done by perchloric acid digestion and starch content was determined spectrophotometrically at 630 nm wavelength.

### 4.5. Gene Expression by RT-PCR Analysis

Total RNA was extracted from young panicles of WT and *osros1a* mutants using TRIzol^®^ Reagent (ThermoFisher, Scientific, Waltham, MA USA). RT reagent Kit PrimeScript^TM^ having gDNA Eraser (Takara, Japan) was used to eliminate genomic DNA residues, then cDNA was synthesized. For RT-PCR reactions, the first strand of cDNA was used with gene-specific primers and the *OsACTIN* gene was used as an internal control (Appendix A).

### 4.6. DNA Methylation Analysis

For bisulfite sequencing of *OsPKS2* and *TDR* gene promoters, genomic DNA from young panicles was extracted by the CTAB method. The EpiTect Bisulfite Kit from Qiagen was used for bisulfite conversion of genomic DNA following the manufacturer’s instructions. The primers were designed using Methyl Primer Express v1.0 (Applied biosystem) (Appendix A). DNA samples were amplified by PCR and separated by 1.5% agarose gel. Then, PCR products were purified with the gel purification kit, and then cloned into the pMD19-T vector and sequenced. Kismeth tool (http://katahdin.mssm.edu/kismeth/revpage.pl) (accessed on 21 October 2019) was used for DNA methylation analysis.

## Figures and Tables

**Figure 1 ijms-23-11349-f001:**
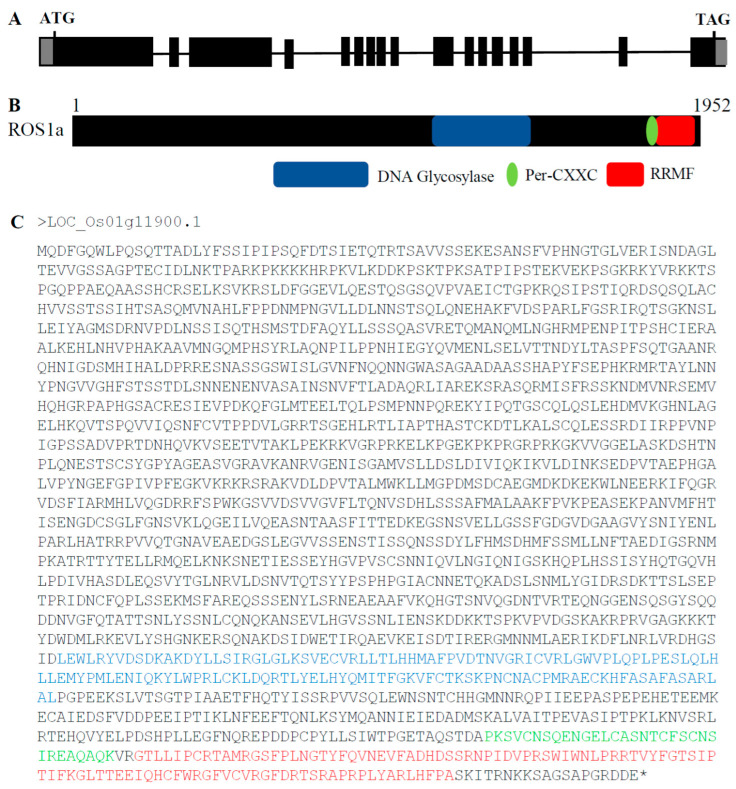
Schematic of *OsROS1a* gene structure and protein features. (**A**) Exons and introns of the *OsROS1a* gene are denoted as black blocks and lines, respectively. The translation initiation codon (ATG) and termination codon (TAG) are shown. (**B**) The protein structure of the *OsROS1a* gene. The open reading frame (ORF) is 1952 amino acid residues. (**C**) The protein sequences of the *OsROS1a* gene. The DNA glycosylase, Per-CXXC and RRMF domains are shown in blue, green and red colors, and all others showed in black.

**Figure 2 ijms-23-11349-f002:**
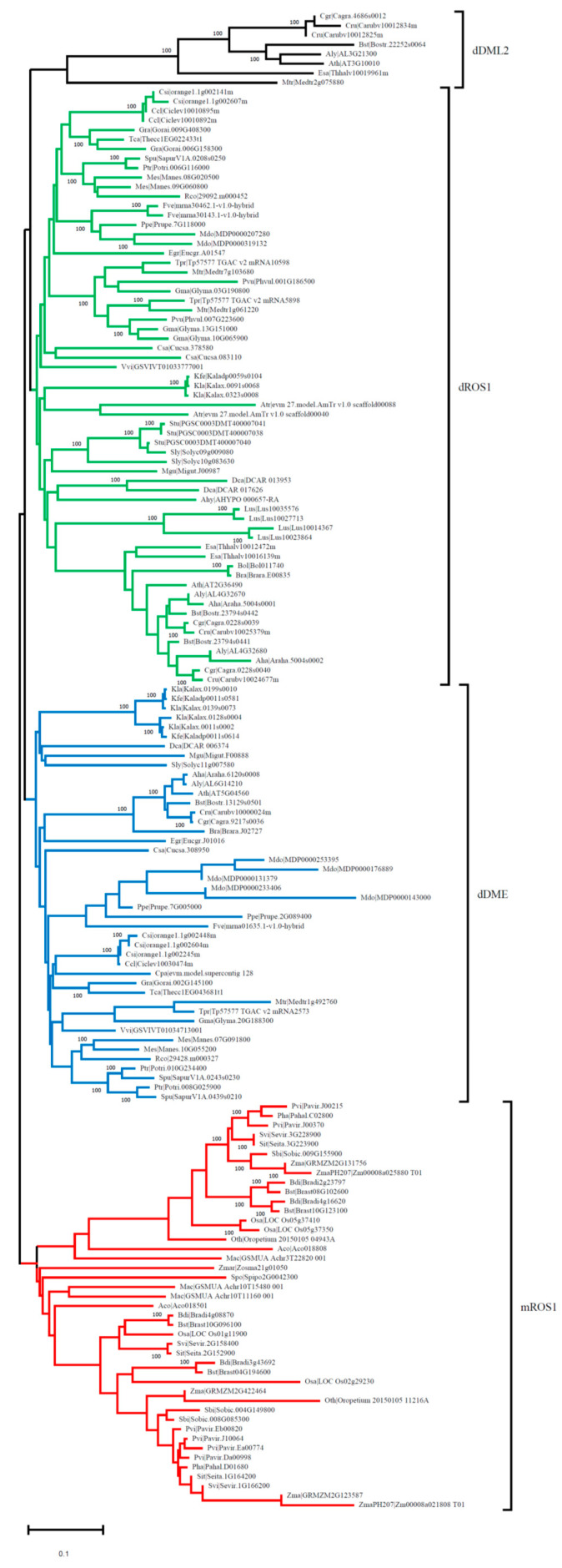
Phylogenetic analysis of the *ROS1* and *ROS1*-like DNA glycosylase family. A phylogenetic tree based on conserved DNA glycosylase domain was constructed. Monocot ROS1 (mROS1) proteins are shown in red color, while dicot DME (dDME), dicot ROS1 (dROS1) and dicot DML2 (dDML2) proteins are shown in blue, green and black, respectively. Protein sequences were downloaded from Phytozome (https://phytozome-next.jgi.doe.gov) (accessed on 19 October 2018).

**Figure 3 ijms-23-11349-f003:**
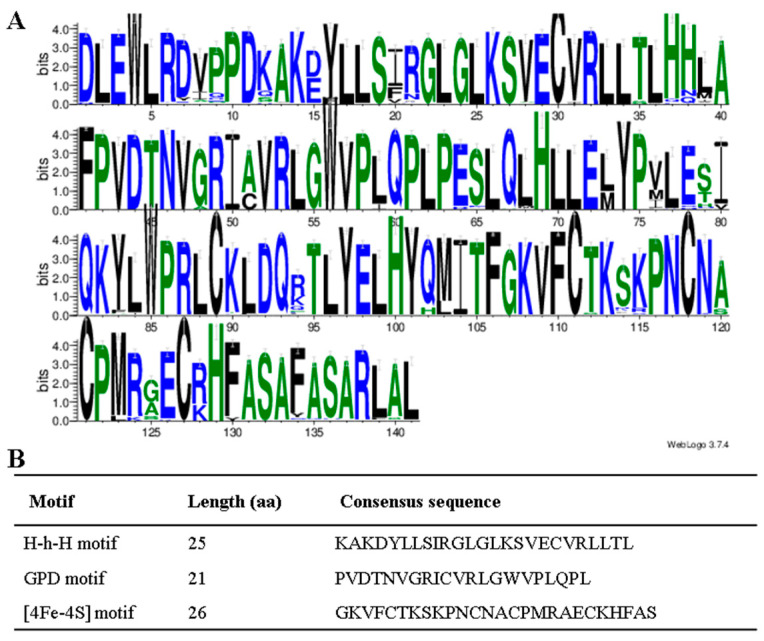
Conserved domain analysis of DNA glycosylase. (**A**) The conserved domain analysis of DNA glycosylase by the WebLogo program. The letter height shows the amino acid residue at each position designating the conservation degree. On the *x*-axis, numbers represent the sequence position in the corresponding conserved domains. The *y*-axis indicates the information content calculated in bits. (**B**) The consensus sequences of three motifs in the glycosylase domain.

**Figure 4 ijms-23-11349-f004:**
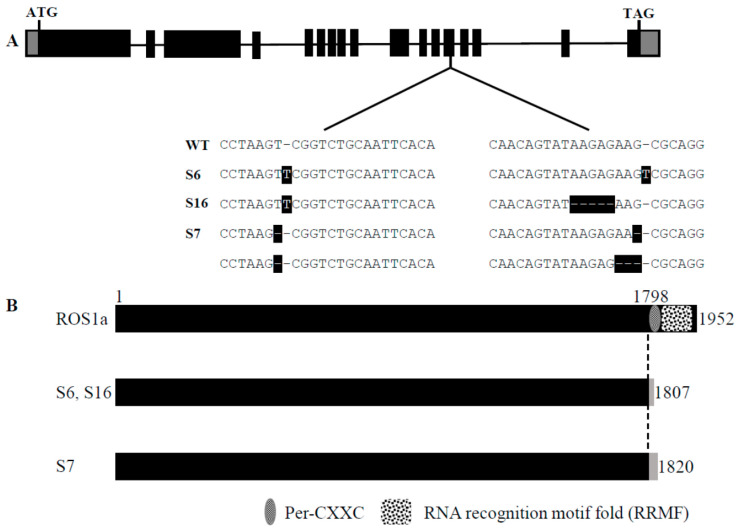
CRISPR/Cas9-induced *OsROS1a* gene modification in rice. (**A**) Schematic of *OsROS1a* gene structure. Exons and introns are denoted as black blocks and lines, respectively. The translation initiation codon (ATG) and termination codon (TAG) are shown. The recovered mutated alleles are shown below the wild-type reference sequence. The target sites nucleotides are indicated in black capital letters. The white dashes with black backgrounds indicate the deleted nucleotides. The white capital letters with black backgrounds indicate the inserted nucleotides. (**B**) Alignment of the protein sequence of *ROS1a* in mutant and wild type. The open reading frame (ORF) is 1952 amino acid residues in length in wild type. The dotted line indicates the frame shift mutation position because of the insertion or deletion of nucleotides due to the gene editing in mutants. The gray color indicates the positions where the premature stop codon was generated in the mutant alleles.

**Figure 5 ijms-23-11349-f005:**
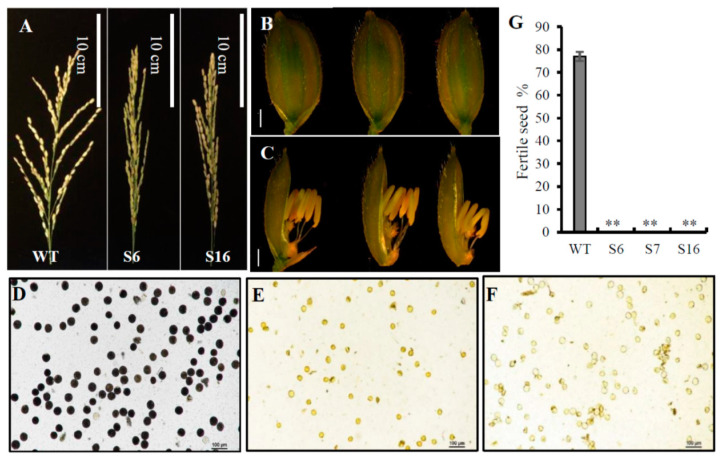
The male fertility phenotype of WT and the *osros1a* mutants. (**A**) Comparison of the mature panicles between WT and *osros1a* mutants showed that *osros1a* mutants failed to produce seeds. Scale bars 10 cm. Comparison of (**B**) the spikelets and (**C**) the anthers phenotype between WT and *osros1a* mutants before anthesis. Scale bars 1 mm. (**D**–**F**) The pollen grains of WT and *osros1a* mutants staining with I_2_-KI staining, respectively. Compared to (**D**) WT, (**E**) S6, and (**F**) S16 failed to form viable pollen. Magnified 20X. (**G**) The fertile seed percentage in WT and S6, S7, S16 mutants. ** denote significant differences from WT at *p* < 0.01 level.

**Figure 6 ijms-23-11349-f006:**
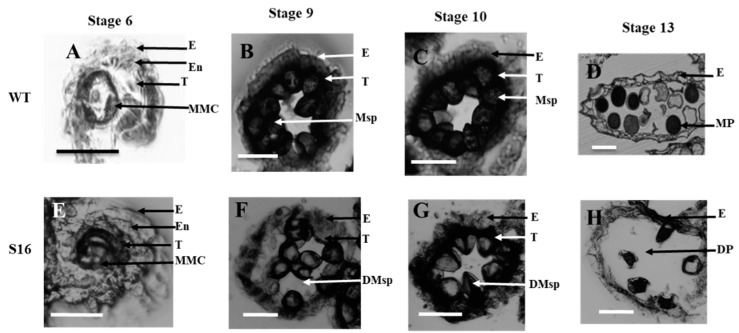
Transverse section analysis of the anther development in WT and *osros1a* S16 mutant. Locules from the anther section of WT and S16 at (**A**,**E**) stage 6 (the microspore mother cells stage), (**B**,**F**) stage 9 (the young microspore stage), (**C**,**G**) stage 10 (the vacuolated pollen stage), and (**D**,**H**) stage 13 (the mature pollen stage). WT sections are shown in (**A**–**D**); *osros1a* S16 mutant in (**E**–**H**). E, epidermis; T, tapetum; En, endothecium; MMC, microspore mother cells; Msp, microspores; MP, mature pollen; DMsp, degenerated microspores; DP, the degenerated pollen. Scale bar: 50 μm.

**Figure 7 ijms-23-11349-f007:**
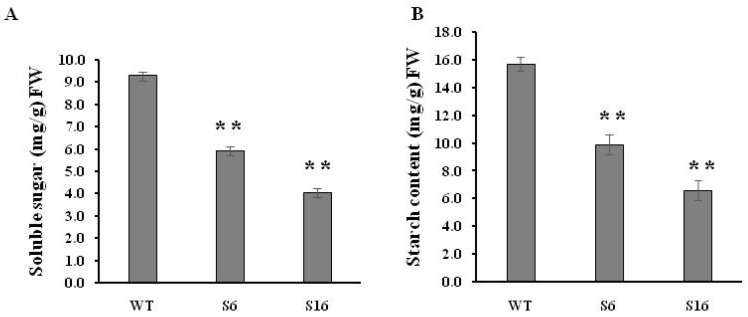
Soluble sugar and starch contents in mature anthers of WT and *osros1a* mutants. (**A**) Total soluble sugar content in mature anthers of WT and *osros1a* mutants. (**B**) Starch content of mature anthers in WT and *osros1a* mutants. Anthers were sampled at stage 13 (the mature pollen stage). Data are means ± SD (*n* = 3). ** denote significant differences from WT at *p* < 0.01 level.

**Figure 8 ijms-23-11349-f008:**
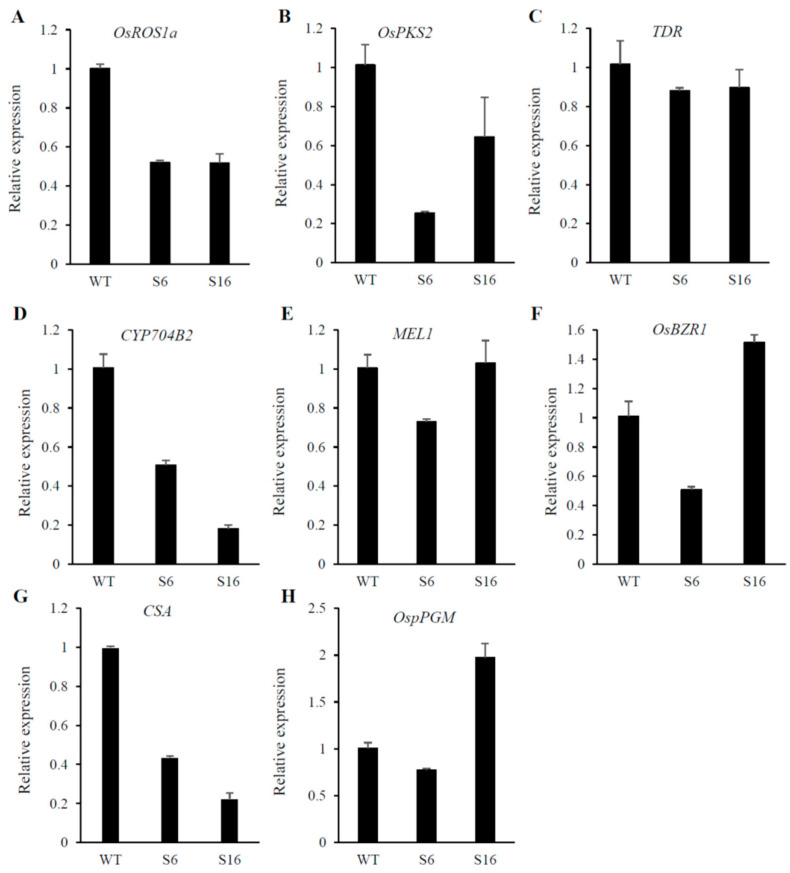
Expression patterns of anther and pollen development-related genes (**A**–**E**) and starch synthesis-related genes were analyzed by qRT-PCR. The young panicles (at stage 10, the vacuolated pollen stage) of WT, *osros1a* S6 and S16 mutants were sampled for RNA extraction. Data are shown as means ± standard deviations, and *OsACTIN* was used as an internal control.

**Figure 9 ijms-23-11349-f009:**
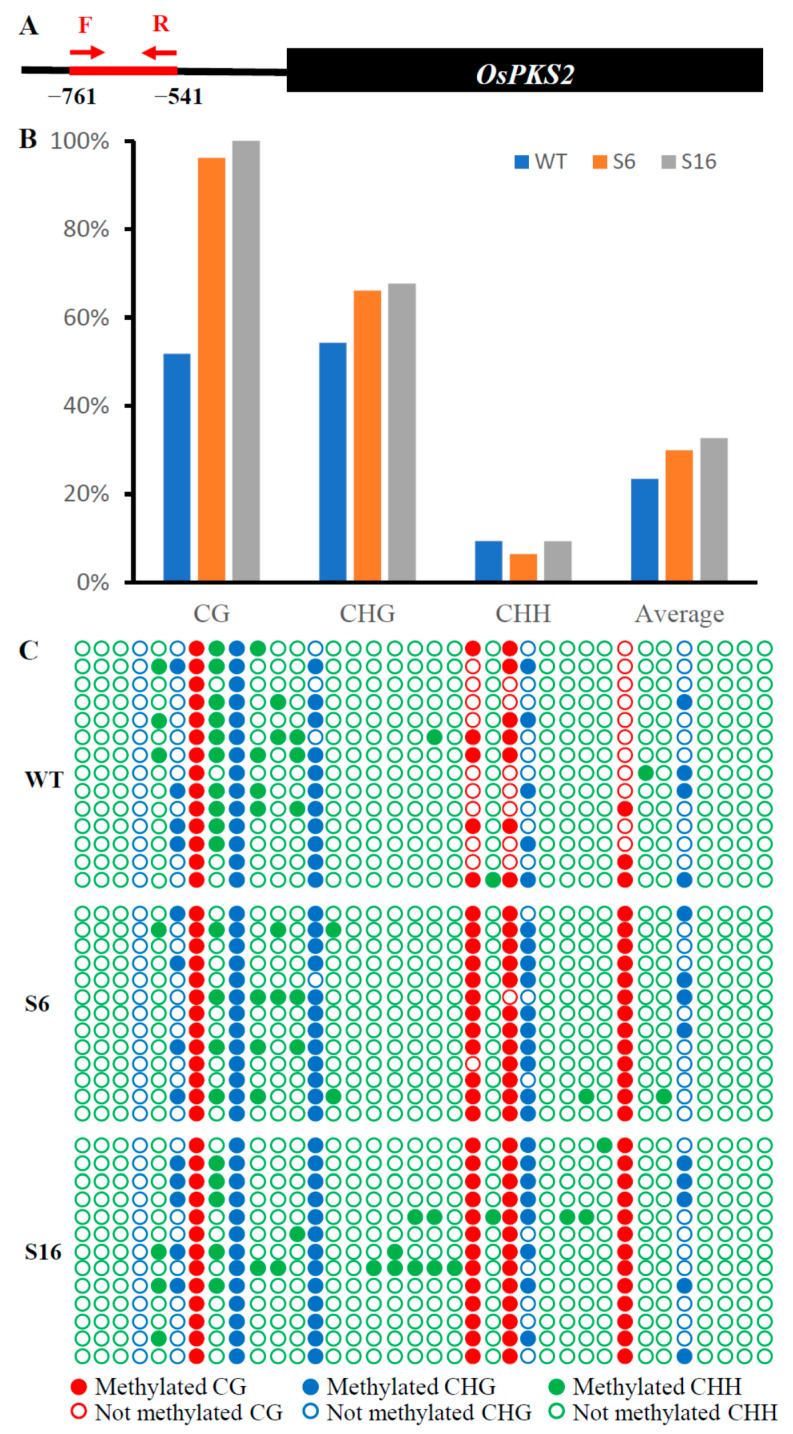
The DNA methylation of *OsPKS2* gene promoter in young panicles. (**A**) The diagram of bisulfite sequencing of 221 bp *OsPKS2* promoter. (**B**) The statistics of bisulfite sequencing of 221 bp *OsPKS2* promoter in WT and *osros1a* mutants. (**C**) The dot plot comparison of DNA methylation between WT and *osros1a* mutants.

## Data Availability

Not applicable.

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
