# Peer review of "Loss of Function of the RRMF Domain in OsROS1a Causes Sterility in Rice (*Oryza sativa* L.)"

_ijms, 2022, doi:10.3390/ijms231911349_

Round 1
Reviewer 1 Report
The authors has done s great job.However my suggestions to authors are:
1.Please add the future perspective of the study.
2.Add limitations of the study.
3. Recheck the manuscript for grammatical mistakes.
Author Response
The authors has done a great job. However my suggestions to authors are:
1.Please add the future perspective of the study.
Response: Thanks very much for the reviewer’s comments. It is a very good suggestion. The future perspective has been added in the revised version.
2.Add limitations of the study.
Response: Very good suggestion. We have added the limitations of the study in the discussion section.
- Recheck the manuscript for grammatical mistakes.
Response: Thanks very much for the reviewer’s constructive suggestion. We carefully went through the whole manuscript and revised the grammatical mistakes, and the native speaker expert also revised and edit the manuscript for us, we think it should be suitable for publication in the journal “International Journal of Molecular Sciences”.
Reviewer 2 Report
The study by Xu et al. attempted to investigate the role of OsROS1a in the reproduction of rice. They results claim that OsROS1a loss of function caused sterility in rice. These results were attributed to the mutation in the RRMF domain of OsROS1a induced employing gene editing technique. The concept of the study creates some interest. However, the study suffers from several missing data and poor description of the results. The methodology is poorly described as well. In addition, the authors should consider revising for English.
Major comments
Although pollen viability test can provide some insight, these results must be supported by other determinant phenotypic traits that are directly affected when a plant is sterile were not evaluated, such as grain weight or panicle weight and grain filling ratio. The authors could have also evaluated the number of spikelet per panicle and the number of panicle per plant considering that all tillers do not produce panicles.
The starch metabolism is crucial during grain filling. I failed to see soluble sugar contents that is more informative than what was presented in the study.
qPCR results are required to see whether the target gene was knocked out or knocked down compared with the WT
In the materials and methods, several important details are missing concerning the gene editing procedure, including the selection and validation of mutant lines for the target genetic locus. This include the gel electrophoresis images, the Agrobacterium transformation process and selection of positive transformants and regeneration. This will allow reproducibility of the results.
Minor comments
See in the attached PDF version of the manuscript

Author Response
The study by Xu et al. attempted to investigate the role of OsROS1a in the reproduction of rice. They results claim that OsROS1a loss of function caused sterility in rice. These results were attributed to the mutation in the RRMF domain of OsROS1a induced employing gene editing technique. The concept of the study creates some interest. However, the study suffers from several missing data and poor description of the results. The methodology is poorly described as well. In addition, the authors should consider revising for English.
Response: Thanks very much for the reviewer’s comments. In the revised version, we have added the qRT-PCR results. The description of results and the methods were carefully modified to make it clear as the reviewer suggested. The native speaker expert revised and edit the manuscript for us, and we think it is suitable to be published in the journal.
Major comments
Although pollen viability test can provide some insight, these results must be supported by other determinant phenotypic traits that are directly affected when a plant is sterile were not evaluated, such as grain weight or panicle weight and grain filling ratio. The authors could have also evaluated the number of spikelet per panicle and the number of panicle per plant considering that all tillers do not produce panicles.
Response: Good suggestion. We investigated the fertile seed percentage, and all osros1a mutants are 0% when compared to 77% in WT, which was added in the revised version. We actually didn’t evaluate the number of spikelet per panicle and the number of panicle per plant, because all other agronomic traits are very similar and the mutants are completely sterile.
The starch metabolism is crucial during grain filling. I failed to see soluble sugar contents that is more informative than what was presented in the study.
Response: Good points. The starch metabolism is crucial during grain filling, and it is very important to analyze the soluble sugar contents for filling grains. While our samples were collected at stage 13 of the anther development, we think it should be fine to do the analysis of the total soluble sugar content for the pollens. We will analyze the soluble sugar contents for our other mutants that is partial sterile and affects the seed shape to reveal the function of OsROS1a gene in the null mutant.
qPCR results are required to see whether the target gene was knocked out or knocked down compared with the WT
Response: Thanks to the reviewer’s comment, qPCR results have been added in the revised version, which improved our manuscript a lot to be suitable for publication and thanks again.
In the materials and methods, several important details are missing concerning the gene editing procedure, including the selection and validation of mutant lines for the target genetic locus. This include the gel electrophoresis images, the Agrobacterium transformation process and selection of positive transformants and regeneration. This will allow reproducibility of the results.
Response: Good points, we have carefully revised the section on materials and methods, and now the missing details have been added in the revised version, thank you very much.
Minor comments
See in the attached PDF version of the manuscript
- it would be appropriate to use base instead of residue
Response: Good point, we have used base instead of residue in the revised version.
- Rephrase for more clarity
Response: Thanks the reviewer’s suggestion, we revised the sentence to make it clear. Replaced “Loss function of ROS1 mutations” as “The loss function of ROS1”.
- An orthologous gene is a gene in different species that evolved from a common ancestor by speciation. The appropriate terminology in this case is paralogs.
Response: Thanks the reviewer for pointing out our mistake, yes it should be paralogs.
- What do you mean? Do you really consider seed production as a trait? This statement sounds not correct. Consider revising to convey a clear and scientific meaning
Response: Thanks to the reviewer’s comment, the sentence has been re-wrote to make it clear and scientific in the revised version.
- The authors should define a clear hypothesis that may be verified or rejected or accepted by the results. Use a common wording such as ..This study aimed..
Response: Great, good suggestion. We revised this part as the reviewer suggested to define a clear hypothesis.
- what do you mean by the longest gene? Do you consider the full gene structure including introns and exons, UTR? Does the number of exons determine the size of a gene? Otherwise, revise and make a reasonable statement.
Response: Thanks to the reviewer’s comment. We have deleted the inappropriate statement.
- Homologue is a general term that include ortholog and paralog. Determine the number of orthologs and paralogs if applicable.
Response: Thanks to the reviewer’s comment. Homologue is a general term that include ortholog and paralog, as the same species has the paralog gene copies and it is difficult to say it is orthologs or paralogs, therefore we used the homologous.
- Add name of species to the phylogenetic tree and all relevant information. This blind tree has no insight.
Response: As the reviewer suggested, we have added the name of species and all relevant information including gene names, thank you very much.
- This section is core to this study but it is poorly described. A thorough and concise description is required.
Response: Thanks the reviewer’s constructive suggestion. we have carefully revised this section, and we think the description of this section is more thorough and concise in the revised version.
- To what these results apply? Are these results compared between WT and mutants? Did the authors recorded 100% sterility in the mutant lines?
Response: Thanks to the reviewer’s comment. Yes, these results were compared between WT and mutants. It is 100% sterility in the mutant lines, and the fertile seed percentage in the mutants is 0% that is showed in Figure 5G in the revised version.
- Improve the caption and present relevant information from the Figure 5
Response: Thanks very much for the reviewer’s comments. We improved the caption and presented relevant information in Figure 5 as the reviewer suggested.
- There is no need to mix statistical difference of alpha between lines for the parameters considered. Use stars or lettering not both.
Response: Thanks for the reviewer pointing out our mistake. We revised this figure and use asterisks to indicate significant differences.
- The authors should analyze soluble sugars not as total but independently.
Response: Good points. It is very important to analyze the soluble sugars independently but not the total for analyzing the filled seeds. While our samples were collected at stage 13 of the anther development, which we think it should be fine to do the analysis of the total soluble sugars for the pollens. We will analyze the soluble sugars independently for our other mutants that is partial sterile and affects the seed shape to reveal the function of OsROS1a gene in the null mutant.
- Indicate the number of days after heading at which samples were collected.
Response: Thanks very much for the reviewer’s comments. Samples were collected at stage 13 of anther development for both Starch and soluble sugar content analysis.
- If you consider the asterisks to indicate the p-value, no need to use the letters on top of bars
Response: Yes, we used the asterisks to indicate the p-value, thank you very much.
- This section is poorly described
Response: Thanks very much for the reviewer’s comments. We carefully re-wrote this section.
- How did the authors select these genes? The authors should provide more information on their possible interaction with the target gene using protein-protein interaction prediction, promoter analysis and qPCR.
Response: Good points. We selected these genes that are commonly used by the published papers. Actually, the best way to do this work is RNA-Seq that will be our future work. As OsROS1a is a gene encodes demethylase that makes DNA demethylation, we prioritized its demethylation function. The reviewer provided us a good direction for further work to investigate protein-protein interaction, which will provide a new insight for revealing the function of OsROS1a gene, many thanks to the reviewer.
- qPCR validation is required to support these results in both WT and mutant lines
Response: Thanks very much for the reviewer’s comments. The qPCR results were added in the revised version.
- On which basis the authors selected these genes when they are other important genes in rice regulating the starch metabolism?
Response: Good points. We predicted that CSA could be the important gene to regulate the sterile phenotype and CSA and related genes were selected. The RNA-Seq and whole-genome bisulfite sequencing will be the target work for the future.
- This section should be explained in detail to allow reproducibility of the results. This should include validation steps of the mutants and gel eletrophoresis images, agrobacterium transformation procedure and sequence validation as well as transformation into plant.
Response: Thanks very much for the reviewer’s comments. We revised this section as the reviewer suggested.